There are amendments to this paper

# The chromosomal organization of horizontal gene transfer in bacteria

Pedro H. Oliveira [1,2], Marie Touchon[1,2], Jean Cury [1,2] & Eduardo P.C. Rocha[1,2]

Bacterial adaptation is accelerated by the acquisition of novel traits through horizontal gene transfer, but the integration of these genes affects genome organization. We found that transferred genes are concentrated in only ~1% of the chromosomal regions (hotspots) in 80 bacterial species. This concentration increases with genome size and with the rate of transfer. Hotspots diversify by rapid gene turnover; their chromosomal distribution depends on local contexts (neighboring core genes), and content in mobile genetic elements. Hotspots concentrate most changes in gene repertoires, reduce the trade-off between genome diversification and organization, and should be treasure troves of strain-specific adaptive genes. Most mobile genetic elements and antibiotic resistance genes are in hotspots, but many hotspots lack recognizable mobile genetic elements and exhibit frequent homologous recombination at flanking core genes. Overrepresentation of hotspots with fewer mobile genetic elements in naturally transformable bacteria suggests that homologous recombination and horizontal gene transfer are tightly linked in genome evolution.

[1] Microbial Evolutionary Genomics, Institut Pasteur, 25–28 rue du Docteur Roux, Paris 75015, France. [2] CNRS, UMR3525, 25–28 rue du Docteur Roux, Paris 75015, France. Pedro H. Oliveira and Marie Touchon contributed equally to this work. Correspondence and requests for materials should be addressed to P.H.O.(email: pcphco@gmail.com) or to M.T.(email: mtouchon@pasteur.fr)

The gene repertoires of bacterial species are often very diverse, which is central to bacterial adaption to changing environments, new ecological niches, and co-evolving eukaryotic hosts[1]. Novel genes arise in bacterial genomes mostly by horizontal gene transfer (HGT)[2], a pervasive evolutionary process that spreads genes between, eventually very distant, bacterial lineages[3]. It is commonly thought that the majority of genes acquired by HGT are neutral or deleterious and thus rapidly lost[4]. Yet, HGT is also responsible for the acquisition of many adaptive traits, including antibiotic resistance in nosocomials[5]. Hence, genome diversification is shaped by the balancing processes of gene acquisition and loss[6], moderated by positive selection on some genes, and purifying selection on many others[7].

Chromosomes are organized to favor the interactions of DNA with the cellular machinery[8]. For example, most bacterial genes are co-transcribed in operons, leading to strong and highly conserved genetic linkage between neighboring genes[9]. At a more global level, early-replicating regions are enriched in highly expressed genes in fast-growing bacteria to enjoy replication-associated gene dosage, creating a negative gradient of expression along the axis from the origin (ori) to the terminus (ter) of replication (ori->ter)[10, 11]. These organizational traits can be disrupted by the integration of novel genetic information. At a local level, new genes rarely integrate within an operon and, instead, they tend to be incorporated at its edges, where they are less likely to affect gene expression[12]. At the genome level, the frequency of integration of prophages in the genome of *Escherichia coli* increases along the ori->ter axis[13]. The results of these studies suggest that the fitness effects of HGT in terms of chromosome organization depend on the specific site of integration.

In prokaryotes, HGT takes place by three main mechanisms: natural transformation, conjugation, and transduction. Mobile genetic elements (MGEs) play a key role in HGT because they are responsible for the latter two processes, respectively by the activity of conjugative elements and phages[14]. Integrative conjugative elements (ICEs) and prophages are large genetic elements that may account for a significant fraction of the bacterial genome[15, 16], and bring to the chromosome many genes in a single event of integration. For example, some strains of *E. coli* have up to 18 prophages[17], and *Mesorhizobium loti*

encodes one ~500 kb ICE[18]. The integration of these large MGEs changes the chromosome size and may split adaptive genetic structures such as operons. This might contribute to explain why most integrative MGEs use site-specific recombinases (integrases) that target very specific sites in the chromosome[19]. Integrases and MGEs have co-evolved with the host genome to decrease the fitness cost of their integration[13].

MGEs carrying similar integrases tend to integrate at the same sites in the chromosome, leading to regions with unexpectedly high frequency of MGEs at homologous regions. This concentration of MGEs in few sites has been frequently described[20, 21], especially in relation to the presence of neighboring tRNA and tmRNA genes[22]. Yet, a previous work described the existence of regions with high rates of diversification in *E. coli* (hotspots), some of which lacked recognizable integrases[23]. In particular, the genes flanking two hotspots were associated with high rates of homologous recombination (*rfb* and *leuX*). In *Streptococcus pneumoniae*, the chromosomal genes flanking MGEs also showed higher rates of homologous recombination[24, 25]. In this species, it was suggested that integration of MGEs close to core genes under selection for diversification could be adaptive by facilitating the transfer and subsequent recombination of the latter[26].

Here, we define and identify hotspots in a large and diverse panel of bacterial species and show how they reflect the mechanisms driving genome diversification by HGT.

## Results

**Quantification of HGT and definition of hotspots.** To study the distribution of gene families in bacterial chromosomes, we analyzed 932 complete genomes of 80 bacterial species (Supplementary Data set 1). We inferred the core genome, the pan-genome, the accessory genome (genes from the pan-genome absent from the core), and the phylogeny of each species, as before[27] (Methods, Supplementary Figs. 1 and 2). We partitioned the genomes into an array of core genes and intervals (Fig. 1, Table 1). The latter were defined as the positions between consecutive core genes. We defined a spot as the set of intervals delimited by members of the same two families of core genes in the genomes of the clade (see Methods for rigorous definitions, Supplementary Fig. 2a, b). We observed that 99.4% of the intervals were part of the species' spots and only 0.6% were in

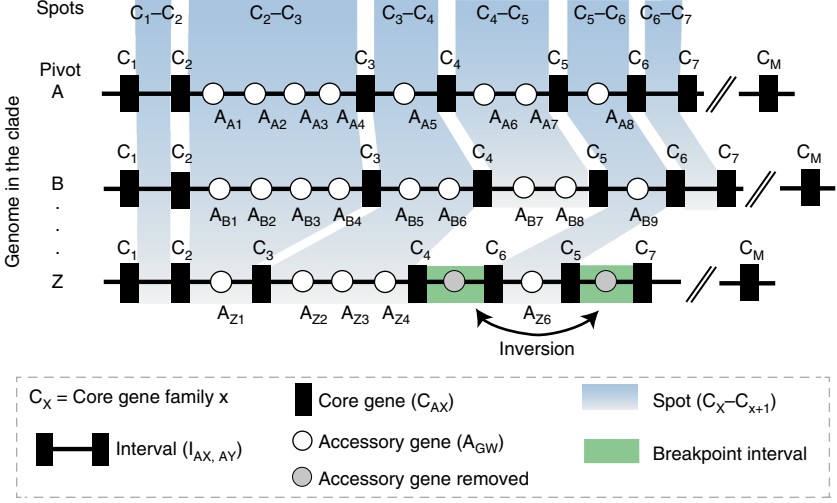

**Fig. 1** Scheme depicting key concepts used in this study. Intervals flanked by the same core gene families (C$_X$, C$_Y$) as those from pivot genome A were defined as syntenic intervals (i.e., the members of the core gene families X and Y were also contiguous in the pivot). The intervals that do not satisfy this constraint were classed as breakpoint intervals (*green-shaded* regions) and excluded from our analysis. For every interval in the pivot genome, we defined spot as the set of intervals flanked by members of the same core gene families (*blue-shaded* regions)

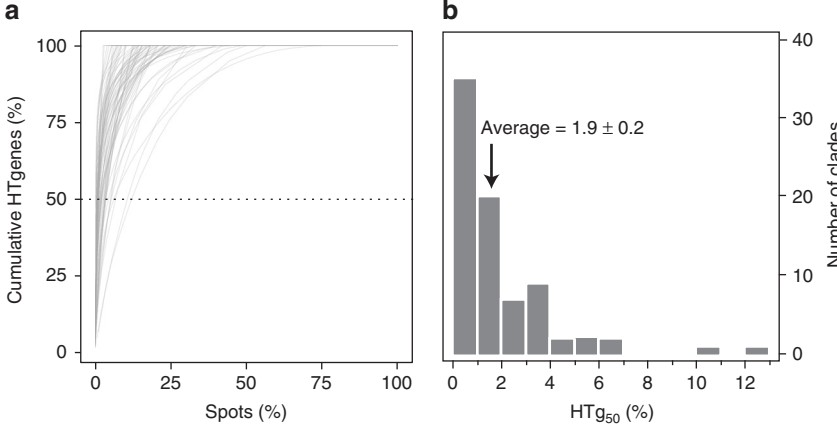

**Fig. 2** Cumulative frequency of HTgenes. **a** Cumulative distribution of horizontally transferred genes (HTgenes, %) in spots for the 80 bacterial clades. **b** Histogram of the minimum number of spots needed to attain 50% of the total number of HTgenes (HTg$_{50}$ index). The average HTg$_{50}$ was only 1.9% (±0.2; standard deviation)

| Table 1 Acronyms used in this study | |
|---|---|
| MGE | Mobile genetic element (i.e., prophage, ICE, IME and integron) |
| ICE | Integrative conjugative element |
| IME | Integrative mobilizable element |
| MAP | Mobility-associated protein (i.e., integrase and transposase (IS)) |
| ARG | Antibiotic resistance gene |
| HGT | Horizontal gene transfer |
| HTgenes | Genes having been horizontally transferred |
| HTg$_{50}$ | Number of spots required to include 50% of HTgenes |
| T$_{95\%}$ | Minimal number of HTgenes required to define a hotspot |

breakpoint intervals. Since 99.8% of spots are flanked by the same two families of core genes in at least half of the genomes of each clade (and 99% in all genomes), it is most parsimonious to consider that the two core genes were already contiguous in the last common ancestor of the clade. Hence, we split the pan-genomes in spot pan-genomes, i.e., sets of gene families located in each spot (Methods, Supplementary Fig. 2). The genes outside spots, i.e., in intervals that were split by events of rearrangement, accounted for < 2% of the total number of genes and were discarded from further analysis.

We used birth-and-death models to identify HGT events in the clade's phylogenetic trees from the patterns of presence/absence of each gene family (Methods). Note that HGT events are defined gene per gene (which will be called HTgenes for Horizontally Transferred Genes), not as blocks, because there are no tools available for the latter and because the goal of our work was to study the clustering of genes acquired by HGT without using a priori models. Spots contained 170,041 HTgenes (15.5% of the total number of accessory genes). We quantified the clustering of these genes by counting the minimal number of spots required to accumulate at least 50% of the HTgenes (HTg$_{50}$) (Fig. 2a). The distribution of these values was skewed toward small values (Fig. 2b). Hence, < 2% of the largest hotspots accumulate >50% of all HTgenes. Conversely, 72.6% of the spots were on average empty, i.e., had no accessory gene in any genome. Similar qualitative conclusions were obtained in the analysis of the distribution of all accessory genes, despite the latter being slightly less clustered (Supplementary Fig. 3). These results show that most HTgenes are integrated in a very small number of sites in the genome.

We used simulations to infer the statistical thresholds for the degree of clustering of HTgenes in each clade (Methods, Supplementary Fig. 4). We made the null hypothesis that these genes are organized in operons like the other genes, and are uniformly distributed among spots. We identified the spot with the highest number of HTgenes in each simulation (Max$_{HTg,i}$), and computed the 95th percentile of the distribution of these maximal values (T$_{95\%}$, Supplementary Data set 1). Simulations disregarding the existence of operons produced lower values of T$_{95\%}$ showing the importance of incorporating information about genetic organization in the model (Supplementary Fig. 5). Spots with more than T$_{95\%}$ HTgenes were called hotspots, spots lacking accessory genes were called empty, and the others were called coldspots. We found a total of 1841 hotspots in the 80 clades (Supplementary Data set 1). They represent only 1.2% of the spots, but they concentrate 47% of the accessory gene families and 60% of the HTgenes.

The number of hotspots differed widely among clades, from none or very few in *Acetobacter pasteurianus*, *Bacillus anthracis*, and the obligatory endosymbionts, to more than 60 in *Bacillus thuringiensis*, *E. coli*, and *Pseudomonas putida* (Fig. 3a). This variance was partly a function of chromosome size (Fig. 3b), but was especially associated with the number of HTgenes (Fig. 3c). Increases in the latter resulted in a less-than-linearly increase in the number of hotspots and in a linear increase in hotspot density per Mb (Supplementary Fig. 6). Hence, a few hotspots aggregate most of the genes acquired by horizontal transfer and this trend is more pronounced when the rates of transfer are high.

**Functional and genetic characterization of hotspots.** We investigated the function of the genes in the spots, using the eggNOG categories, to assess if hotspots were enriched in particular traits (Methods, Fig. 4a). Genes classified as poorly characterized or as having an unknown function were not considered in the subsequent functional analyses (they were 13.1% of the total). We then compared the distribution of the functions of all accessory genes and that of HTgenes in hotspots relative to coldspots. Both analyses showed an underrepresentation of translation and post-translational modification genes in hotspots. These genes tend to be essential and are less frequently transferred horizontally[28]. In contrast, hotspots overrepresented genes associated with cell motility, defense mechanisms, transcription, and replication and repair. Moreover, around 9% of the hotspots encoded antibiotic

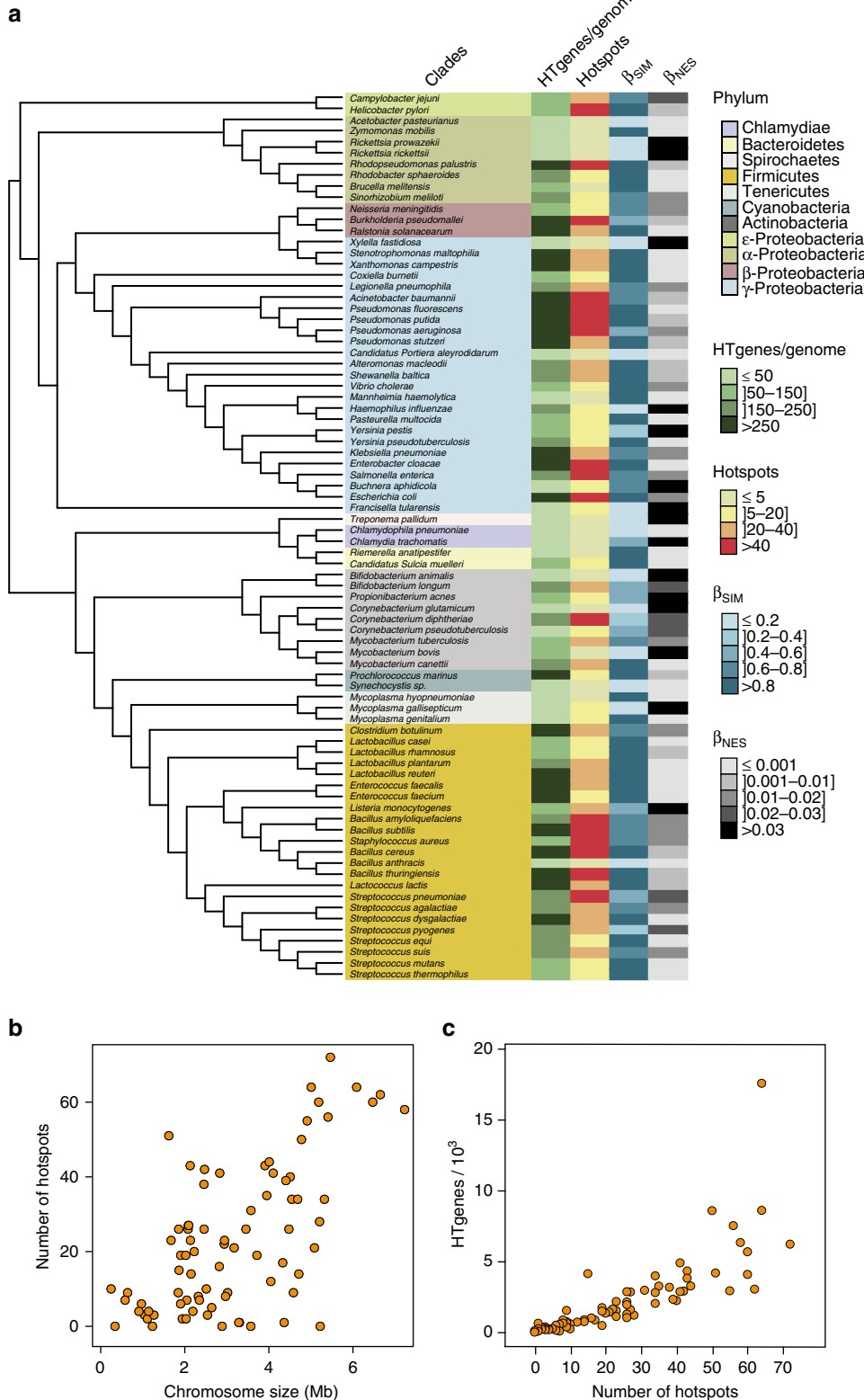

**Fig. 3** Analysis of HTgenes and the abundance and distribution of hotspots. **a** 16S rRNA phylogenetic tree of the 80 bacterial clades. The tree was drawn using the iTOL server (itol.embl.de/index.shtml)[70]. The first *column* indicates the clade and is *colored* by phylum. The four subsequent *columns* correspond respectively to: the average number of HTgenes per genome computed using Count, the number of hotspots, the average Simpson dissimilarity index ($\beta_{SIM}$, accounting for turnover), and the average multiple-site dissimilarity index accounting only for nestedness ($\beta_{NES}$). These values are given in Supplementary Data set 1. **b** Distribution of the average number of hotspots per clade according to the average genome size ($G_S$). **c** Association between the number of hotspots and the number of HTgenes in the clade

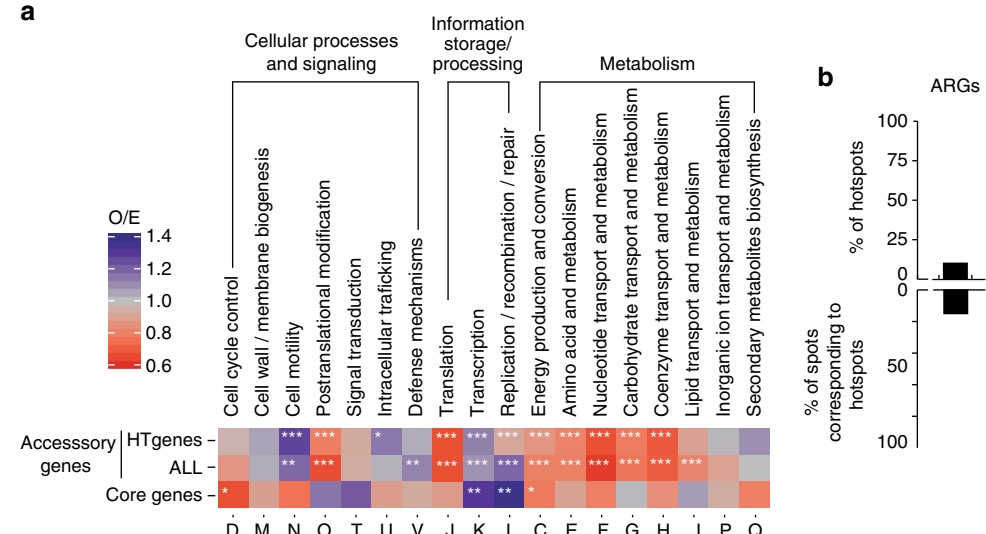

**Fig. 4** Functional characterization of hotspots. **a** Observed/expected (O/E) ratios of non-supervised orthologous groups (NOGs, shown as *capitalized letters*). The first two *lines* represent the values of HTgenes and accessory genes observed in hotspots when the null model was computed from the distribution of the same type of genes in coldspots. The last *line* shows the same type of analysis for the core genes flanking hotspots when the null model is computed using the core genes not flanking hotspots. Expected values were obtained by multiplying the number of HTgenes, accessory, or core genes in hotspots by the fraction of genes assigned to each NOG. *$P < 0.05$; **$P < 10^{-2}$; ***$P < 10^{-3}$, $\chi^2$-test. **b** Percentage of hotspots with antibiotic resistance genes (ARGs, *top*), and percentage of spots with ARGs that are hotspots (*bottom*). Note that hotspots are only 1.2% of all the spots

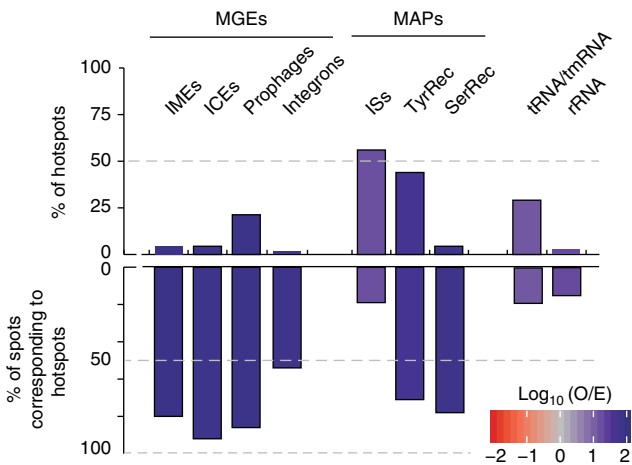

**Fig. 5** Genetic mobility of hotspots. We represent the percentage of hotspots containing the different genetic elements (*top*) and the percentage of spots containing such elements that are hotspots (*bottom*). Note that hotspots are only 1.2% of all the spots. The analysis includes MGEs (IMEs, ICEs, prophages, integrons), mobility-associated proteins (MAPs) (ISs, TyrRec, SerRec), and tRNA/tmRNA, rRNA. Also, shown in *colored bins* are the observed/expected ($\log_{10}$O/E) number of hotspots that contain the abovementioned elements, when the null model was computed from the distribution of coldspots containing the same type of elements. Expected values were obtained by multiplying the number of hotspots by the fraction of spots containing each type of element

resistance genes (ARGs), which is much more than expected by chance (0.8%) (Fig. 4b).

Some of the functions overrepresented in hotspots—defense, replication, repair—are typically found in MGEs, which concentrate in specific loci targeted by integrases (often at tRNAs). Accordingly, the vast majority of self-mobilizable MGEs—89% of the prophages and 90% of ICEs—were identified in hotspots (Supplementary Data set 3 and Supplementary Fig. 7). On the

other hand, only around 9% of the hotspots encoded ICEs or integrative mobilizable elements (IMEs), and only 23% encoded prophages (Fig. 5). Integrons were even rarer (present in 1% of the hotspots). Non-self-transferable MGEs lack conjugation or virion structural genes, but usually encode integrases. The vast majority of integrases was identified in hotspots, but less than half (45%) of the hotspots encoded an integrase and only 29% encoded tRNA or tmRNA genes (Fig. 5). Hence, although most self-mobilizable MGEs are in hotspots, most hotspots lack them (Supplementary Fig. 8).

Insertion sequences (ISs) encoding DDE recombinases (transposases) are frequent within MGEs, and we found them in many hotspots (56%). The integration of these elements has low-sequence specificity, which explains why hotspots accounted for a small fraction of the locations with ISs (19%), unlike what we observed for self-mobilizable MGEs and integrases. Altogether, half of the hotspots lacked evidence for the presence of MGEs and 27% lacked any of the mobility-associated proteins (MAPs, integrases and transposases) that we searched for. These results confirm that hotspots concentrate most MGEs and integrases, but not the majority of ISs. They also show that regions with high concentration of HTgenes often lack recognizable MGEs, suggesting that other mechanisms are implicated in their genesis and turnover.

**The chromosomal context of hotspots.** We then searched to identify the preferential genetic contexts of hotspots, since they might illuminate constraints associated with the chromosomal organization of HGT. We analyzed whether the distribution of hotspots was random relative to the function of the neighboring core genes. Interestingly, these core genes showed an overrepresentation of several functions, notably replication, recombination/repair, and transcription (Fig. 4a). In contrast, cell cycle control genes were underrepresented. Hence, hotspots are preferentially associated with specific functions of neighboring core genes.

We then tested whether hotspots were randomly distributed in genomes. Since replication drives much of the large-scale

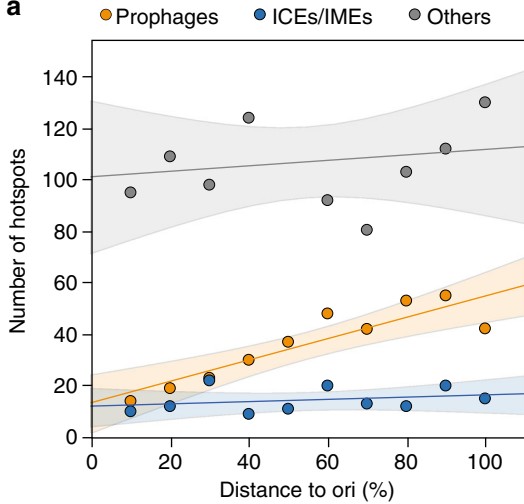

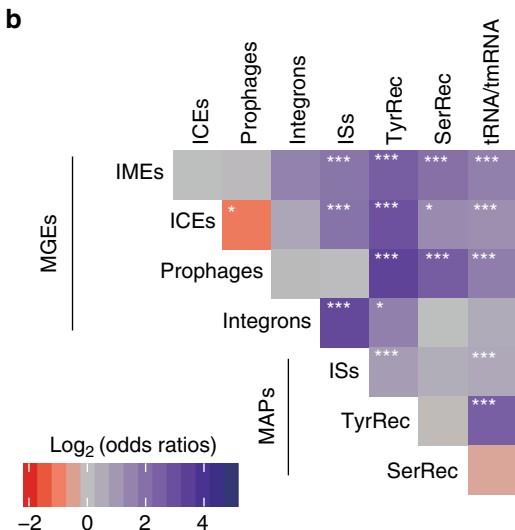

**Fig. 6** Chromosomal context of hotspots. **a** Number of hotspots containing prophages, ICEs/IMEs, and none of the above along the origin–terminus axis of replication. Linear regression and the confidence limits (*shaded area*) for the expected value (mean) were indicated for each category. The number of hotspots including prophages increases linearly with the distance to the origin of replication (Spearman's $\rho = 0.87$, $P < 10^{-3}$), but this is not the case for the other two categories (both $P > 0.05$). **b** Heatmap of odds ratios of co-localizations in hotspots of MGEs, mobility-associated proteins (MAPs) and RNAs. ***$P < 10^{-3}$; **$P < 10^{-2}$; *$P < 0.05$; Fisher's exact test

organization of bacterial genomes[8], we analyzed the position of hotspots relative to the distance to the origin of replication along the replichore. These results showed that the frequency of hotspots including prophages, as previously shown in *E. coli*[13], increases linearly along the ori-> ter replication axis (Fig. 6a). Interestingly, this does not seem to be the case for ICEs and IMEs, nor for the very large category of hotspots that lack ICEs, IMEs, and prophages.

As these results show that prophages and ICEs have different distribution patterns, we quantified the frequency of co-occurrence of different MGEs and MAPs in the same hotspots (but not necessarily in the same intervals, Fig. 6b). In line with expectations, most MGEs significantly co-occurred with integrases, integrons, ISs, and tRNAs. The most notable exceptions concerned the prophages, that did not significantly

co-occurred with ISs, presumably because ISs are rare in phages[29], or integrons, and they were found less frequently than expected in spots with conjugative elements. This is in line with the analysis showing that they have specifically different distributions along the chromosome replication axis.

**Genetic diversity of hotspots**. The integration of a MGE in the chromosome adds a large number of genes in one single location, potentially creating a hotspot on itself. Such events result in a concentration of HTgenes in a genome (strain-specific integration), or in several genomes (when the integration took place at the last common ancestor of several strains). The distribution of the number of genomes with orthologous HTgenes in hotspots suggests that these cases are relatively rare (Supplementary Fig. 9a). Only 8% of the hotspots had all accessory gene families represented in one genome (Supplementary Fig. 9b, c). Hence, few of these regions seem to have been created by the integration of a single MGE.

To assess whether genetic diversity in hotspots was compatible with one single ancient integration event, we introduced measures derived from the analysis of beta diversity in Ecology, where it is used to measure the differences in species composition between different locations[30] (Methods). Here we used it to measure the difference in gene repertoires among a set of intervals from the same spot. We measured the Sørensen index ($\beta_{SOR}$) for hotspots and coldspots of each species using the binary matrix of gene presence/absence. Diversity results from a mixture of independent gene acquisitions and replacements (turnover) and differential gene loss (nestedness), and $\beta_{SOR}$ can be partitioned into the two related additive terms: turnover ($\beta_{SIM}$) and nestedness ($\beta_{NES}$) ($\beta_{SOR} = \beta_{NES} + \beta_{SIM}$, Fig. 7a).

Beta diversity of accessory genes was higher in hotspots than in coldspots (Fig. 7b). This difference was caused by turnover, since only $\beta_{SIM}$ was significantly higher in hotspots than in coldspots (Fig. 7c). The values of $\beta_{NES}$ were very low in both cases; confirming that most hotspots are not caused by singular events of integration of MGEs. We obtained similar results when the analysis of diversity was restricted to HTgenes (Supplementary Fig. 10). While genetic diversity is high in hotspots and coldspots, these results show faster diversification in hotspots because they endure higher genetic turnover.

Finally, we wished to test whether hotspots lacking MAPs had such a high genetic turnover that MGEs would be rapidly removed. We split the hotspots into two categories: hotspots containing and lacking MAPs. Both categories showed values of genetic diversity close to one that were caused by high turnover. Nevertheless, hotspots lacking MAPs showed slightly lower values for these variables (Supplementary Fig. 11). Hence, the absence of MAPs in these hotspots is not due to an excess of genetic turnover.

**Hotspots of homologous recombination**. Many hotspots lack identifiable MGEs or even integrases. Yet, they show high genetic diversity, suggesting that other mechanisms may drive their evolution. We tested the possibility that these regions could integrate HTgenes by homologous recombination at the flanking core genes, as suggested for certain hotspots of *E. coli*[23] and *S. pneumoniae*[24] (Fig. 8a). Our hypothesis predicts higher levels of homologous recombination in core genes flanking hotspots than in the rest of the core genome. We tested this prediction in two complementary ways. Firstly, we detected homologous recombination events in the core genes using ClonalFrameML (Methods). We found 50% more recombination events in core genes flanking hotspots than in the other core genes (Fig. 8b). Secondly, we searched for evidence of phylogenetic incongruence between each

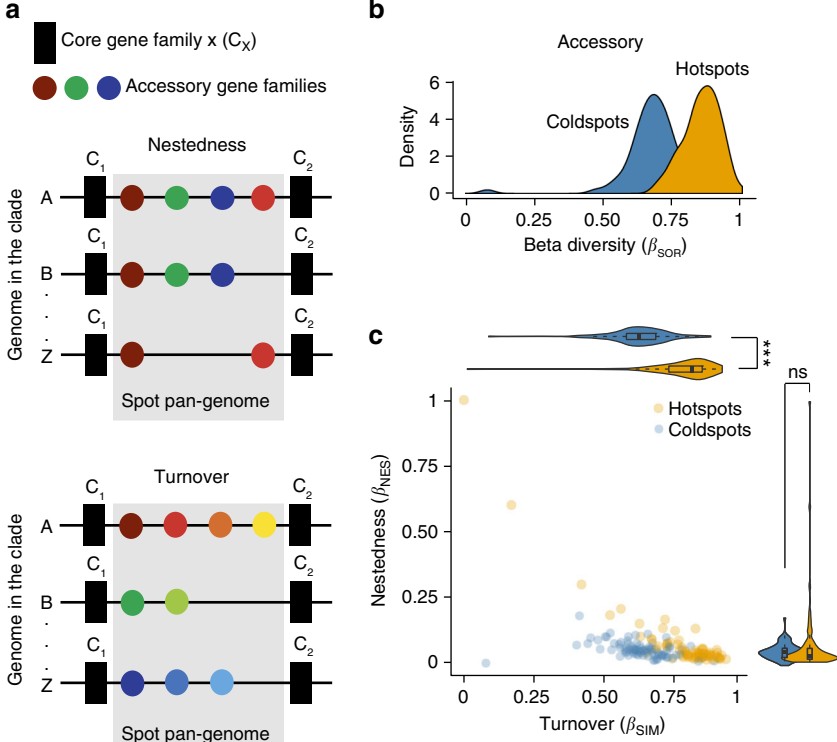

**Fig. 7** Genetic diversity of the accessory genes present in hotspots and coldspots. **a** Examples of gene nestedness and turnover in a spot. Turnover measures the segregation between intervals in terms of gene families, i.e., it accounts for the replacement of some genes by others. Nestedness accounts for differential gene loss and measures how the gene repertoires of some intervals are a subset of the repertoires of the others. It typically reflects a non-random process of gene loss. **b** Distributions of $\beta$ diversity ($\beta_{SOR}$) in hotspots and coldspots. **c** Partition of $\beta_{SOR}$ in its components of nestedness ($\beta_{NES}$) and turnover ($\beta_{SIM}$) for hotspots and coldspots ($\beta_{SOR} = \beta_{NES} + \beta_{SIM}$). ***$P < 10^{-3}$; Mann–Whitney–Wilcoxon test; ns: not significant

core gene family and the whole core genome tree of the clade using the Shimodaira–Hasegawa (SH) test (Methods). The number of genes with significant phylogenetic incongruence was 30% higher among core genes flanking hotspots than among the others (Fig. 8b). In line with these observations, core genes flanking hotspots also had higher nucleotide diversity (Fig. 8c). We found qualitatively similar results when the analysis was performed on a per species or per genus basis (Supplementary Data set 5). Hence, core genes flanking hotspots are more targeted by recombination processes than the others.

Naturally transformable bacteria have the ability to acquire genetic material independently of MGEs. In these species, transfer of chromosomal material mediated by homologous recombination at the flanking core genes might be particularly frequent. To test this hypothesis, we put apart the 19 bacterial species that are known to be naturally transformable in our dataset[31] (Supplementary Data set 1). We observed that these species had more hotspots than the others ($P < 0.05$, Mann–Whitney–Wilcoxon test). We searched for MAPs in these hotspots and observed that they also had fewer hotspots with MAPs ($P < 0.05$, Mann–Whitney–Wilcoxon test). Finally, recombination was 20% more frequent in core genes flanking hotspots in naturally transformable than in the remaining bacteria ($P < 10^{-4}$; $\chi^2$-test). These results suggest that recombination at core genes flanking hotspots might be particularly important in driving genetic diversification of naturally transformable bacteria.

## Discussion
Our study showed high concentration of HTgenes in a small number of locations in the chromosomes of many bacterial species. These hotspots include most MGE-related genes, fitting

previous observations that the latter co-evolved with the host to use integrases targeting specific locations in the chromosome that minimize the fitness cost of chromosomal integration. For example, many temperate phages integrate tRNA genes without disrupting their function[32]. The concentration of most self-mobilizable MGEs at few loci might be thought sufficient to justify the existence of hotspots, but we found that few hotspots had identifiable prophages or conjugative elements and that most lacked integrases. These puzzling results could be caused by failure to identify MGEs, but our methods were shown to be highly accurate at identifying conjugative elements and prophages[13, 33], or by the presence of many radically novel integrase-lacking MGEs in these model microbial species, which would be very surprising. Hotspots also contain degenerate MGEs that we have failed to identify. Yet, inactivated elements are not expected to drive the observed rapid genetic turnover of these regions.

Our results suggest that an MGE-independent mechanism, double homologous recombination at the flanking core genes, contributes to hotspot diversification. The mechanism only requires housekeeping recombination functions and exogenous DNA with homology to the flanking core genes. This last condition is easy to fulfill, because these genes are present in all genomes of the species (and usually in closely related species). In agreement with our hypothesis, we showed that naturally transformable species had more hotspots, and fewer MAPs in hotspots, than the others. There are other mechanisms of transfer that can bring homologous sequences without MAPs in non-transformable bacteria, including generalized transduction, gene transfer agents, or DNA-carrying vesicles[34]. Their role in hotspot diversification remains to be explored.

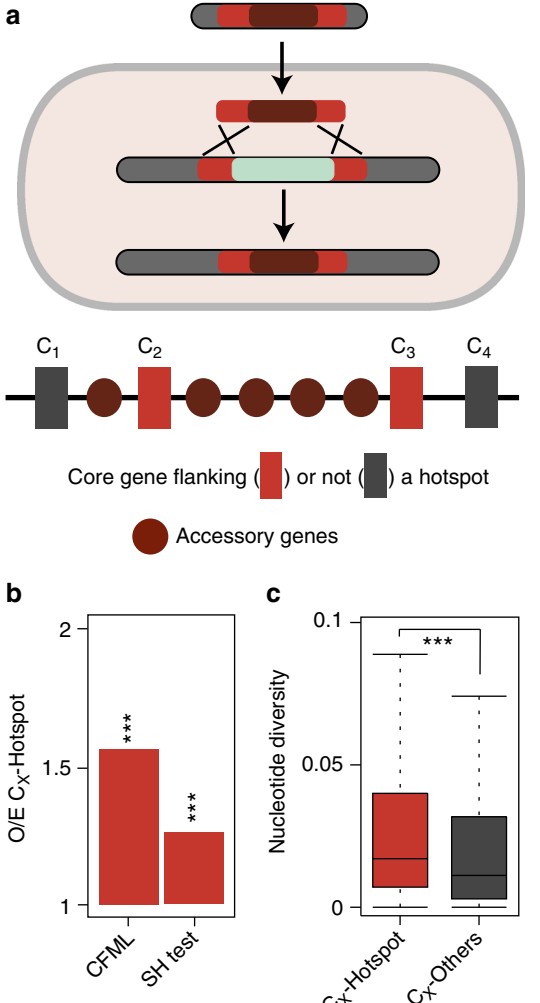

**Fig. 8** Evidence for more frequent homologous recombination in core genes flanking hotspots than in the other core genes. **a** Model for the creation and evolution of hotspots by homologous recombination at the flanking core genes. **b** We detected homologous recombination events in the core genes using ClonalFrameML, and searched for evidence of phylogenetic incongruence ($P < 0.05$) between each core gene family and the whole core genome tree of the clade using the Shimodaira–Hasegawa (SH) test. The observed–expected ratios (O/E) for these two analyses are significantly higher than one. ***$P < 10^{-3}$; $\chi^2$-test. **c** Differences in nucleotide diversity between core genes flanking hotspots and the others. Nucleotide diversity was calculated using the R package "PopGenome" v2.1.6[49] by implementation of the diversity.stats() command. ***$P < 10^{-3}$; Mann–Whitney–Wilcoxon test

Many HTgenes are not adaptive (or even deleterious) and are rapidly lost by genetic drift (or purifying selection)[6, 7, 35]. Nevertheless, regions of high concentration of HTgenes must also include adaptive genes, as shown here for ARGs. In these circumstances, the high genetic turnover at hotspots might seem paradoxical, because it may lead to their loss. Actually, even adaptive genes can be lost with little fitness cost under certain circumstances. Genes under diversifying selection, such as defense systems, may be adaptive for short periods of time and subsequently lost (or replaced by analogous genes)[36]. Some costly genes may be adaptive in only very specific conditions, such as ARGs[37], and become deleterious for the cell fitness upon environmental change. Finally, some genes under frequency-dependent selection, such as toxins[38], may stop being adaptive when their frequency changes in the population. Genetic drift, purifying, diversifying, and frequency-dependent selection can thus contribute to the rapid turnover of HTgenes. As a consequence of their high genetic turnover, hotspots are expected to be enriched in genes of specific adaptive value.

Hotspots may affect bacterial fitness not only by the genes they contain, but also by the way they drive genome diversification. According to the chromosome-curing model[39], hotspots may facilitate the elimination of elements with deleterious fitness effects, such as certain MGEs, by double recombination at the flanking core genes. This fits our observation that core genes flanking hotspots endure higher rates of homologous recombination. As a response to chromosome curing, natural selection is expected to favor MGEs that inactivate genes encoding recombination and repair proteins[39]. Interestingly, we also found that hotspots tend to be flanked by recombination and repair core genes. Although these genes seem intact, at least they respect the constraints that we imposed for their classification as core genes, their expression may be affected by HTgenes in the neighboring hotspot. For example, excision of a MGE in *Vibrio spendidus* 12B01 from a *mutS* gene downregulates the expression of the latter leading to a hypermutator phenotype[40].

Several selective effects can contribute to explain the very different number of hotspots per species, which were strongly correlated with the number of HTgenes and weakly with genome size (itself also correlated with the rate of HGT[27]). The first association may explain why species with little genetic diversity, such as *B. anthracis* and mycobacteria, have few hotspots in spite of their large genome size. It is also possible that our statistical tests lack power when species have few HTgenes. Some ecological determinants also affect the number of HTgenes, and their concentration in the genome. For example, sexually isolated species with few MGEs, such as obligatory endosymbionts, are expected to have few hotspots. Many of these species may also inefficiently select for hotspots because they have low effective population sizes. Conversely, the highest number of hotspots was found in facultative pathogens with very diverse gene repertoires, including *E. coli*, *Pseudomonas spp.*, and *Bacillus cereus*. A rigorous statistical assessment of the ecological traits affecting the organization of HTgenes will require the analysis of a larger panel of species representative of the different prokaryotic lifestyles.

Overall, our results suggest that hotspots are the result of the interplay of several recombination mechanisms and natural selection, presumably because they minimize disruption of genome organization by circumscribing gene flux to a small number of permissive chromosomal locations. For example, the increase in prophage-containing hotspots along the ori-> ter axis suggests co-evolution between these elements and the host to remove prophages from early replicating regions that are also rich in highly expressed genes in fast growing bacteria[13]. Interestingly, the spatial distribution of the remaining hotspots does not show similar patterns, which can be due to the lower fitness costs associated with their excision. Further work is needed to understand if there are other organizational traits that constrain the distribution of hotspots in the chromosome, and in particular in those devoid of recognizable MGEs. Knowing these traits might facilitate large-scale genetic engineering and should lead to a better understanding of the evolutionary interactions between horizontal gene transfer and genome organization.

Finally, our study focused on the dynamics of hotspots and how they contribute to genome diversification, but left unanswered the questions related to their origin and fate. Previous studies identified common prophage hotspots between *E. coli* and *Salmonella enterica*[13]. Hence, we will have to study

taxonomical units broader than the species level to unravel their origin. As for their fate, long-term adaptive HTgenes may become fixed in the population, explaining the patterns of nestedness of certain hotspots, and leading eventually to the split of the hotspot into two new (eventually hot) spots.

## Methods

**Data**. The sequences and annotations of 932 bacterial genomes from 80 bacterial species were retrieved from Genbank Refseq (ftp://ftp.ncbi.nih.gov/genomes, last accessed in February 2014)[41]. We made no selection on the species that were to be analyzed, except that we required a minimum of four complete genomes per species. We have made no attempt to re-define species: we used the information presented in GenBank. Their list is available in Supplementary Data set 1. We excluded CDS annotated as partial genes, as well as those lacking a stop codon or having stop codons within the reading frame. Core genomes and phylogenetic reconstructions were obtained from our previous work[27] (Supplementary Data set 2). Our data set includes several species from the same genera. It also includes species with diverse numbers of genomes and HTgenes. To minimize the effects of these unavoidable biases most of our analyses are non-parametric and each species has the same weight. When they were done on the data cumulated from all species, we made a control where each species is analyzed separately. We also made complementary analyses where we aggregated the results per genus. The references for these supplementary controls are indicated in the main text, and the data are in the Supplementary Material.

**Identification of core genomes**. We used 80 core genomes previously published[27]. These core genomes were built for clades with at least four complete genomes available in GenBank RefSeq (Supplementary Data set 1, Supplementary Fig. 1). Briefly, a preliminary list of orthologs was identified as reciprocal best hits using end-gap-free global alignment, between the proteome of a reference genome (pivot, typically the first completely sequenced isolate) and each of the other strain's proteomes. Hits with < 80% similarity in amino-acid sequence or >20% difference in protein length were discarded. This list of orthologs was then refined for every pairwise comparison using information on the conservation of gene neighborhood. Thus, positional orthologs were defined as bi-directional best hits adjacent to at least four other pairs of bi-directional best hits within a neighborhood of 10 genes (five upstream and five downstream). These parameters (four genes being less than half of the diameter of the neighborhood) allow retrieving orthologs at the edge of rearrangement breakpoints (positions where intervals were split by events of chromosome rearrangement) and therefore render the analysis robust to the presence of a few rearrangements. The core genome of each clade was defined as the intersection of pairwise lists of positional orthologs.

**Definitions of interval and spot**. The core genome is the collection of all gene families present in one and only one copy in each genome of a clade (Supplementary Fig. 2). Let $C_X$ and $C_Y$ be two families of core genes in a clade with N taxa where one of the taxa is a pivot (reference genome, see above). We call $C_{AX}$ and $C_{AY}$ contiguous core genes in a given chromosome A if they are adjacent in the list of core genes sorted in terms of the position in the chromosome. We defined an interval ($I_{AX, AY}$) as the location between the pair of contiguous core genes $C_{AX}$ and $C_{AY}$ in chromosome A. The content of an interval is the set of accessory genes in the interval. The HTgenes content of an interval is the number of genes that were acquired by HGT in the interval. Multiple chromosomes, when present, were treated independently.

Intervals flanked by the same core gene families ($C_X$, $C_Y$) as the pivot genome were defined as syntenic intervals (i.e., the members of the core gene families X and Y were also contiguous in the pivot). The intervals that do not satisfy this constraint were classed as breakpoint intervals and excluded from our analysis. They contain < 2% of all genes. For every interval in the pivot genome, we defined spot as the set of syntenic intervals flanked by members of the same pair of core gene families (Supplementary Fig. 2).

**Identification of spot pan-genomes**. The pan-genome is the full complement of homologous gene families in a clade. We built a pan-genome for each species using the gene repertoire of each genome. Initially, we determined a preliminary list of putative homologous proteins between pairs of genomes (excluding plasmids) by searching for sequence similarity between each pair of proteins with BLASTP v.2.2.28+ (default parameters). We then used the $e$-values (<10$^{-4}$) of the BLASTP output to cluster them using SILIX (v1.2.8, http://lbbe.univ-lyon1.fr/SiLiX)[42]. We set the parameters of SILIX such that two proteins were clustered in the same family if the alignment had at least 80% identity and covered >80% of the smallest protein (options –I 0.8 and –r 0.8). We computed the diversity of gene families observed in each spot. The spot pan-genome is the set of gene families present in the intervals associated with the spot (Supplementary Fig. 2).

**Reconstruction of the evolution of gene repertoires**. We assessed the evolutionary dynamics of gene repertoires of each clade using Count[43] (downloaded in

April 2015). This program uses birth-death models to identify the rates of gene deletion, duplication, and loss in each branch of a phylogenetic tree. We used the spots' pan-genomes matrices, and the phylogenetic birth-and-death model of Count, to evaluate the most likely scenario for the evolution of a given gene family on the clade's tree. Rates were computed with default parameters, assuming a Poisson distribution for the family size at the tree root, and uniform gain, loss, and duplication rates. One hundred rounds of rate optimization were computed with a convergence threshold of 10$^{-3}$. After optimization of the branch-specific parameters of the model, we performed ancestral reconstructions by computing the branch-specific posterior probabilities of evolutionary events, and inferred the gains in the terminal branches of the tree. The posterior probability matrix was converted into a binary matrix of presence/absence of HTgenes using a threshold probability of gain higher than 0.95 at the terminal branches and excluding gains occurring in the last common ancestor with a probability higher than 0.5.

**Identification of hotspots**. We made simulations to obtain the expected distribution of HTgenes in the spots given the numbers of HTgenes and spots (Supplementary Fig. 4). We made the null hypothesis that the distribution of these genes was constrained by the frequency of genes in operons, and followed a uniform distribution in all other respects. Previous works have shown that two-third of the genes are in operons and one-third are in mono-cistronic units[44], with little inter-species variation for the average length of poly-cistronic units ($3.15 \pm 0.06$)[45]. Hence, given N HTgenes per clade we created two groups of elements: N/3 isolated genes and 2 N/3 in operons with three genes. These elements were then randomly placed among the spots following a uniform distribution. For each of the 1000 simulations (per species), we recorded the maximal value of genes within a single spot (Max$_{HTg,i}$), which was used to identify the value of the 95th percentile (T$_{95\%}$) of the distribution of Max$_{HTg,i}$. Hence, 95% of the simulations have no spot with more than T$_{95\%}$ genes (Supplementary Data set 1). Spots (in the real genomes) with more than T$_{95\%}$ HTgenes were regarded as hotspots. Spots lacking accessory genes were called empty spots. The other spots were called coldspots.

As a control, we also made simulations considering that HTgenes were acquired independently of the structure in operons (i.e., considering N isolated genes). The values of T$_{95\%}$ of the two analyses were highly correlated (Spearman's $\rho = 0.89$, $P < 10^{-4}$, Supplementary Data set 1), but those of the latter were smaller (linear regression: T$_{95\% \ isolated} = -0.62 + 0.66 \ T_{95\% \ operons}$, $R^2 = 0.87$). This is expected because the operon structure should increase the variance of the genes per spot, and thus increase T$_{95\%}$.

**Measures of gene repertoire diversity**. Since most spots have few or no genes, and most gene families have few (or no gene) per genome, we computed the genetic diversity of spots using matrices of presence/absence of gene families (computed from the pan-genome).

We computed beta diversity per clade, using a multiple-site version (each interval is the equivalent of a site)[46] of the widely used Sørensen dissimilarity index ($\beta_{SOR}$):

$$\beta_{SOR} = \frac{\sum_{i<j} \min(b_{i,j}, b_{j,i}) + \sum_{i<j} \max(b_{i,j}, b_{j,i})}{2 \cdot (\sum_i S_i - S_T) + \sum_{i<j} \min(b_{i,j}, b_{j,i}) + \sum_{i<j} \max(b_{i,j}, b_{j,i})}, \quad (1)$$

where $S_i$ is the total number of accessory genes in genome $i$, $S_T$ is the total number of accessory genes in all genomes considered together, and $b_{i,j}, b_{j,i}$ are the numbers of accessory genes present in genome $i$ but not in $j$ ($b_{i,j}$) and vice-versa ($b_{j,i}$).

We then used a partitioned version of the ecological concept of beta diversity to characterize the gene diversity of spots[46]. $\beta_{SOR}$ can be partitioned into two additive terms: turnover ($\beta_{SIM}$) and nestedness ($\beta_{NES}$) ($\beta_{SOR} = \beta_{NES} + \beta_{SIM}$, Fig. 7a).

To compute the turnover we used the multiple-site version[46] of the Simpson dissimilarity index ($\beta_{SIM}$):

$$\beta_{SIM} = \frac{\sum_{i<j} \min(b_{i,j}, b_{j,i})}{(\sum_i S_i - S_T) + \sum_{i<j} \min(b_{i,j}, b_{j,i})}, \quad (2)$$

This index is a measure of the evenness with which families of genes are distributed across intervals of a spot (it is a measure of segregation). Turnover implies the replacement of some gene families by others.

By definition, the multiple-site dissimilarity term accounting only for nestedness ($\beta_{NES}$) results from the subtraction[46]:

$$\beta_{NES} = \beta_{SOR} - \beta_{SIM}. \quad (3)$$

Nestedness occurs when intervals with fewer genes are subsets of intervals with larger gene repertoires. It reflects a non-random process of gene loss.

The above formulae were computed as follows: first, we plotted the distribution of the number of accessory genes from the hotspots of all clades analyzed. We took the minimum of this distribution (min$_d$) and used it to select coldspots with a number of accessory genes equal or higher than min$_d$. By doing this, we eliminated coldspots with very few accessory genes, and likely to introduce a bias while computing diversity (leading to extreme situations where $\sum S_i \approx S_i$; $\beta_{SIM} \approx 1$, and as consequence $\beta_{NES} \approx 0$). After this filtering step, we put together all hotspots and all coldspots of each genome in two separate concatenates to avoid statistical artifacts

associated with poorly populated spots. The diversity was computed per clade for each of the concatenates.

**Inference of homologous recombination**. We inferred homologous recombination on the multiple alignments of the core genes of each clade using ClonalFrameML (CFML) v10.7.5[47] with a predefined tree (i.e., the clade's tree), default priors $R/\theta = 10^{-1}$, $1/\delta = 10^{-3}$, and $\nu = 10^{-1}$, and 100 pseudo-bootstrap replicates, as previously suggested[47]. Mean patristic branch lengths were computed with the R package "ape" v3.3[48], and transition/transversion ratios were computed with the R package "PopGenome" v2.1.6[49]. The priors estimated by this mode were used as initialization values to rerun CFML under the "per-branch model" mode with a branch dispersion parameter of 0.1.

**Functional assignment**. Gene functional assignment was performed by searching for protein similarity with HMMer (hmmsearch) on the bactNOG subset of the eggNOG v4.5 database[50] (downloaded in March 2016). We have considered the pivot (reference) genomes as good representatives of each clade, and limited our analysis to these. We have kept hits with an *e*-value lower than $10^{-5}$, a minimum alignment coverage of 50%, and when the majority (>50%) of non-supervised orthologous groups (NOGs) attributed to a given gene pertained to the same functional group. Hits corresponding to poorly characterized or unknown functional groups were discarded.

**Identification of MGEs and proteins associated to mobility**. Temperate phages integrated in the bacterial chromosome (prophages) were identified using Phage Finder v4.6[51] (stringent option). Prophages with > 25% of the predicted genes belonging to ISs, and partially degraded prophages (shorter than 30 kb) were removed[52]. Integrons were identified using IntegronFinder v.1.4 with the –local_max option[53]. Integrative conjugative elements (ICEs) and integrative mobilizable elements (IMEs) were identified using MacSyFinder v.1.0.2[54] with TXSScan profiles[55]. Elements with a full conjugative apparatus were classed as ICE, the others as IME (see ref. 33 for criteria). Integrases were identified using the PFAM profiles PF00589 for tyrosine recombinases, and the pair of profiles PF00239 and PF07508 for serine recombinases (http://pfam.xfam.org/)[56]. All the protein profiles were searched using hmmsearch from the HMMer suite v.3.1b1 (default parameters). Hits were regarded as significant when their *e*-value was smaller than $10^{-3}$ and their alignment covered at least 50% of the protein profile. Insertion sequences (ISs) were detected combining two approaches (i) using hmmsearch from the HMMer with IS HMM profiles (as previously proposed)[57] and (ii) by a BLAST-based method using the ISFinder database[58]. Integrases and transposases were defined as mobility-associated proteins (MAPs). tRNA genes were identified using tRNAscan SE v.1.21[59], tmRNA genes were identified using Aragorn v.1.2.37[60], and the location of the rRNA genes was taken from the Genbank annotation file. Antibiotic resistance genes were detected using HMMer against the curated database of antibiotic resistance protein families ResFams (Core v.1.2, http://www.dantaslab.org/resfams)[61] using the '-cut_ga' option. A hotspot was considered to encode a peculiar MGE or MAP when at least one genome of the clade contained such element (Supplementary Data set 3).

**Identification of origin and terminus of replication**. The ori and ter of replication were predicted using Ori-Finder in the pivot genome of each clade[62]. When the ratio of the predicted replichores length was greater than 1.2, the clade was removed from the analysis (Supplementary Data set 4). Then, we divided each replichore in 10 equally sized regions from the ori to the ter of replication.

**Phylogenetic analyses**. We retrieved the 16S rRNA sequences of the sequenced type strains (also used as reference genomes, see above) of the 80 bacterial clades (Fig. 3a). We made a multiple alignment of them with MAFFT v7.305b[63] using default settings, and removed poorly aligned regions with BMGE v1.12[64] using default settings. The tree was computed by maximum likelihood with PHYML v3.0[65] under the general time reversible (GTR) + Γ(4) + I model (Supplementary Data set 2a). This tree is never used in the calculations; it is only used in Fig. 3a to display the relative position of each clade in the phylogeny of bacteria.

We built core genome trees for each clade using a concatenate of the multiple alignments of the core genes (see main text). Each clade's tree was computed with RAxML v8.00[66] under the GTR model and a gamma correction (GAMMA) for variable evolutionary rates. All trees are shown in Supplementary Data set 2b. We performed 100 bootstrap experiments on the concatenated alignments to assess the robustness of the topology of each clade's tree. The vast majority of nodes were supported with bootstrap values higher than 90% (Supplementary Data set 2b). We inferred the root of each phylogenetic clade's tree using the midpoint-rooting approach of the R package "phangorn" v1.99.14[67]. The alignment and the tree for each individual core gene were used for topology testing against the clade's tree (i.e., the concatenate tree of all the core genes of the clade) using the Shimodaira–Hasegawa (SH) congruence test[68] (1000 replicates) implemented in IQ-Tree v1.4.3[69].

**Data availability**. The authors declare that all data supporting the findings of this study are available within the article and its Supplementary Information files and from the corresponding author upon reasonable request.

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

## Acknowledgements

This work was supported by an European Research Council grant (EVOMOBILOME no. 281605). J.C. is a member of the 'École Doctorale Frontière du Vivant (FdV)—Programme Bettencourt'. We thank the members of the Microbial Evolutionary Genomics group for comments and suggestions on the manuscript.

## Author contributions

P.H.O., M.T. and E.P.C.R. designed the research; P.H.O., M.T. and E.P.C.R. analyzed the data; J.C. provided data and tools; P.H.O., M.T. and E.P.C.R. wrote the paper.

## Additional information

**Competing interests:** The authors declare no competing financial interests.

