## [Peer Review File · Nature Communications]

Reviewers' comments:

Reviewer #1 (Remarks to the Author):

This manuscript makes five major claims: i) that mobile genetic elements (MGEs) are located in hotspots, ii) that most hot spots do not contain MGEs, iii) that core genes flanking hot spots more frequently than other genes, iv) that transformable bacteria have more hotspots and less MGEs than other bacteria and v) that hot spot genes facilitate rapid diversification.

While it is well known that MGEs are concentrated to specific regions of the genome, the other four claims are mostly novel. Although some of the claims, such as for example that genes flanking MGEs show higher recombination rates than other core genes have been made before for individual species, the broad and general approach taken in this manuscript is to my knowledge novel. This paper will be of interest to others in the field and probably influence the thinking about the mechanisms that generate variability within a bacterial population (especially the importance of transformation to the generation of gene content diversity).

The definition of spots, hot spots, empty spots, spot pan-genomes, intervals, breakpoint intervals and nestedness are central to the manuscript and it would be motivated to include a schematic figure at the very beginning of the result section to illustrate how these terms are used in the context of the manuscript. I would also suggest a box or table that summarizes all abbreviations (MGE, ICE, IME, MAP, HTG, HTG50, T95, etc.). While the claims made in the manuscript are supported by numerous statistical analyses for this particular dataset, I am wondering how sensitive the overall estimates and conclusions are to the selection of genomes. For example, some genera like *Pseudomonas* and *Bacillus* contain many species with many hotspots (Figure 2), while other genera are only represented by a single species. This could be tested for example by removing one genus at a time and then redo the calculations. Yet other genera with sequenced genomes seem not to be represented at all. The criteria for inclusion or exclusion of genomes for the analyses should be mentioned.

The methods are well described and enough details are given. The manuscript is overall well written, but a bit difficult to follow due to the many numbers and statistical tests. Some of the numbers could perhaps be moved to a table and summarized. The results are discussed in the context of the previous literature. However, a discussion about the selective advantage associated with different types of hotspots is lacking – what is the biological significance of the patterns observed? Or are the frequencies and different types of hotspots just an effect of random chance?

Reviewer #2 (Remarks to the Author):

I like this paper very much and as expected with these authors, it is a careful and detailed analysis. My overall comment is that the paper might benefit from shortening and tightening up on the editing. I felt some of the explanations were a little overly long.

Specific comments:

Page 3, line 56: might be also interesting to include the paper from Sharp et al on evolutionary rate variation around the genomes of enterobacteria. It was the first of its kind, i think. (Sharp, P.M., Shields, D.C., Wolfe, K.H. and Li, W.-H. (1989) Chromosomal location and evolutionary rate variation in enterobacterial genes. *Science* 246, 808-810.)

Page 5, line 111: spots should be spot

Page 5, line 114: This is a bit of rhetoric. no need to say "state of the art".

Page 7, line 158: relatively should be relative

Page 8, top part of the page. Consider shortening this or tightening the language. I found it very verbose. Would a table help?

Page 8, line 190: change "[...]and 24% others[...]" to "[...]and another 24%[...]"

Page 10: The bet diversity part is not explained well enough, I feel. Could you add a few sentences, perhaps explaining how this works? I understand beta diversity and I think i understand this analysis, but I am not certain the I understand it, so perhaps a bit of help for the reader might be in order?

Page 15, line 353: "Expectedly" is a bit awkward. Perhaps "in line with expectations"?

Reviewer #3 (Remarks to the Author):

This study from Oliveira and collaborators aims to identify recombination hotspots in bacterial genomes and to describe their genomic distribution and origins. Authors analyzed genomic data available for 80 bacterial species to infer recent HTG events and identify mobile genetic elements potentially involved in these events.

Although results from this meta-analysis are convincing and may provide interesting insights about the dynamics of bacterial genomes faced to HGT, I don't think that put in a sufficiently broad evolutionary nor functional context. Especially, I found this study very descriptive, both in the results (where authors intentions' are often missing) and the methods, too focused on technical approaches (e.g. statistical measures of HGT) and the discussion too vague about potential evolutionary implications. Especially, this study is mostly focused on genomic aspects (genome size, composition, structure) and MGE dynamics but very few on functional aspects and bacterial ecology (a bit is however mentioned about antibiotic resistance or endosymbiosis L349-350 but without real discussion). I wanted to see more about what can we say about differences observed among species regarding what we now about their ecology, phylogenetic relations, cell cycle, lifestyle? Furthermore, the way analyses are described makes them very hard to understand at first glance. For instance, most of the very first paragraph (L96-113) from results is confusing without the help from the (very clear!) figure S1; definitions are fuzzy and the corresponding method doesn't help so much (I still didn't get what a "pivot genome" and a "spot pan-genome" are?). I think this is especially a concern for the Nat Com broad audience and scope.

For these reasons I suggest authors to submit their paper maybe to a more specific journal, after deep revision of the manuscript. Please found below more specific comments that I hope will help the authors to improve their work:

ABSTRACT

L30: Please be more clear about the fact that the clustering increases with genome size

L31-33: This sentence is very descriptive and too vague; I know authors tried to summarize very contrasted results but their should put more emphasize on concrete and interesting examples (e.g. the concentration of some classes of MGE in a small number of hotspots) rather than fuzzy generalities (e.g. "local and global chromosomal context").

INTRODUCTION

L48-51: the way authors made the transition is weird. Plus, the fact that most of HTGs that are lost are not adaptive doesn't exclude purifying selection.

L72: What kind of structural changes?

L76: Define DDE

L85-88: This sentence sounds a bit like a circular reasoning, maybe reformulate?

L89: This sentence alone reveals that the purpose of the study is very vague and not put in a sufficiently broad context (see major comments). Also, precise "recombination" hotspot

L93: precise in Bacteria

RESULTS

L96: Provide here an introducing sentence about your purpose with this analysis.

L97: Provide a supplementary figure explaining how you defined core, pan and accessory genomes (not clear from the text)

L100-103: Incomprehensible without the figure. Plus, what is a pivot genome? the closest from the root of a clade? Or was it arbitrary chosen by authors (which is completely fine to me as far as it is clearly indicated). Intervals, spots, breaks points, etc should be more clearly defined directly in the result section.

L110-111: I didn't get what a "spot pan-genome" is, even after reading the method.

L114: This over-use of compound words doesn't help non-initiated readers

L116: Could you provide a much less confusing acronym than "HTGs"? E.g. HTgenes

L154: Provide here an introducing sentence about your purpose with this analysis.

L197-198: Why this statement?

L218-221: The link between the result and the interpretation are not obvious at first glance

L236-248: unless the scope of the paper is purely technical, this part should be in the method, and only briefly described in results

DISCUSSION:

L288: Author should start the discussion putting their study in a broader context. Plus, the number of hotspots is not informative, first because there's no range across species nor statistics given by authors, second because one has no obvious expectation regarding which number it should be. Why is it surprising? What do we expect?

L291: "specific" rather than "certain"?

L295-299: This part should be removed in results or method.

L307: Is it the same result than L289?

L307-310: No real discussion here. I have the feeling that this section is purely filling

L312: please explain more clearly the link you are making between your observation and your interpretation

L314: purifying selection?

L320-321: Is it so surprising regarding the huge size of these classes of MGE? Intuitively if they are greater, I expect those to be found in a smaller number of hotspots.

L322-326: Very naively, could we imagine that MBE dynamics in these regions is so fast (or under very strong purifying selection) that it cannot be detected at the considered evolutionary scales?

L348-350: Could authors dig a bit more about the link between hotspot and lifestyle? Is there any analysis that could be done in this way, knowing species lifestyle?

L348-355: It's very unclear what authors try to point out here. That the association between the number of HGT and hotspot frequency/density is just an artifact of HTG number /genome size association? Please clarify.

L363: "certain organizational pattern" is too vague

L364: again, filling sentence

L370: It could be interesting to use predicted DNA folding sites, if available in bacteria studied by authors, and look if these sites correlate with hotspot distribution in these species.

L374: "certain function", too vague

L376-377: What is the link with the previous sentence and what is the authors' point here?

METHOD:

L383-384: It could be interesting to look backward if hotspots are enriched in these CDS

L418-419: Does it mean that potential HTG that led to these rearrangements were not estimated?

L423-438: This full section is unclear and should be rewritten

L441: evolutionary dynamics?

L459: any reference for this?

L460-461: is this proportion is highly variable among species? If so, could it impact your results?

FIGURES:

Fig1: sulcia is a species name, Buchnera a genus, please uniformize

Fig2a: very descriptive panel. What can we conclude from it? Is there any family effect? Lifestyle?

Fig3a: what the letter at the bottom stand for?

REVIEWERS' COMMENTS:

Reviewer #2 (Remarks to the Author):

This is an exciting paper that provides you with some very interesting and detailed insights into the process of horizontal gene transfer in prokaryotes. The current version is much improved in terms of readability and I have no problems with it. There are still some grammatical and language issues, but they are fairly minor.

Reviewer #3 (Remarks to the Author):

Authors greatly improved their manuscript and clarified most of my queries.

Minor comments:

L90-92: This sentence should be re-written, because the causal link between the observation and the proposed mechanism is not intuitive.

L206: Very naively, the observed trade-off between prophages and integrative elements occurrence in hotspots couldn't be simply due to their large size, and so their potentially large negative effect on fitness? I guess that in the same way, it is also rare to observe several prophages or several ICEs in the same hotspot?

L250-252: Be more specific about "high genetic diversity"

L284-293: The discussion starts too abruptly without general context but with method elements. Maybe the authors should remove the very first paragraph from the discussion, or to summarize it in a single sentence, while putting less (or no) emphasis on the method they used.

Answers to the Reviewers

Please refer to the word version of the revised manuscript, where the location of each answer is highlighted.

Reviewer 1

This manuscript makes five major claims: i) that mobile genetic elements (MGEs) are located in hotspots, ii) that most hot spots do not contain MGEs, iii) that core genes flanking hot spots more frequently than other genes, iv) that transformable bacteria have more hotspots and less MGEs than other bacteria and v) that hot spot genes facilitate rapid diversification.

While it is well known that MGEs are concentrated to specific regions of the genome, the other four claims are mostly novel. Although some of the claims, such as for example that genes flanking MGEs show higher recombination rates than other core genes have been made before for individual species, the broad and general approach taken in this manuscript is to my knowledge novel. This paper will be of interest to others in the field and probably influence the thinking about the mechanisms that generate variability within a bacterial population (especially the importance of transformation to the generation of gene content diversity).

We thank the Reviewer for recognizing the interest and potential impact of our work in the field.

Comment nº 1.1:

The definition of spots, hot spots, empty spots, spot pan-genomes, intervals, breakpoint intervals and nestedness are central to the manuscript and it would be motivated to include a schematic figure at the very beginning of the result section to illustrate how these terms are used in the context of the manuscript. I would also suggest a box or table that summarizes all abbreviations (MGE, ICE, IME, MAP, HTG, HTG50, T95, etc.).

Answer to comment nº 1.1:

We have taken some elements from Supplementary Fig. 4 to elaborate a more detailed Fig. 1. The latter now introduces the reader to several key concepts used in the context of the manuscript. The concept of nestedness appears much later in the text, so we have decided to introduce it only in Fig. 7. As suggested by the Reviewer, we have also added a table with all abbreviations (Table 1).

Comment nº 1.2:

*While the claims made in the manuscript are supported by numerous statistical analyses for this particular dataset, I am wondering how sensitive the overall estimates and conclusions are to the selection of genomes. For example, some genera like *Pseudomonas* and *Bacillus* contain many species with many hotspots (Figure 2), while other genera are only represented by a single species. This could be tested for example by removing one genus at a time and then redo the calculations. Yet other genera with sequenced genomes seem not to be represented at all. The*

criteria for inclusion or exclusion of genomes for the analyses should be mentioned.

Answer to comment nº 1.2:

There are two points in this comment:

1) How sensitive the analysis is to a given species. We should note that in many analyses each species has the same impact (it counts as one observation). In this case, removing one species does not affect the results, especially since we have used non-parametric statistics, which are more robust to outliers. But sometimes, we have just cumulated the data for all species. In this case there is indeed the risk that some species drive the results. We tackled the five key results of our work highlighted above by the Reviewer:

i) That mobile genetic elements (MGEs) are located in hotspots.

We now show histograms (Supplementary Fig. 7a) of the percentage of MGEs in hotspots per species. We made the same analysis where we put together species of the same genus (Supplementary Fig. 7b). This shows that no single species/genus is unduly driving this result. The results are also shown in Supplementary Dataset 3b-c.

ii) That most hotspots do not contain MGEs.

We now show histograms (Supplementary Fig. 8) of the percentage of hotspots lacking MGEs and integrase proteins. The analysis was performed per species and per genus. The results show that no single species/genus is unduly driving this result. The results are also shown in Supplementary Dataset 3d-e.

iii) That core genes flanking hotspots recombine more frequently than other genes.

For each species and genus we computed the values of CFML recombination events for the core genes flanking the hotspots and for the others. We then made pairwise comparisons between the populations using Wilcoxon tests. The results, shown in Supplementary Dataset 5, show that no particular species/genus drives the results. We have now included these observations in the manuscript.

iv) That transformable bacteria have more hotspots, and more hotspots without MGEs than other bacteria.

In the manuscript we did mention already that naturally transformable bacteria have more hotspots than the others ($P < 0.05$, Mann–Whitney–Wilcoxon test). This observation came from the analysis per species, which is also shown in Supplementary Dataset 1. In this particular case we cannot perform the analysis by genus, since we would have situations where a certain genus would simultaneously contain naturally transformable and non-transformable bacteria. In any case, this would ultimately increase the variability inside each genus, and therefore, reduce redundancy. We also checked that transformable bacteria have more hotspots with no MGE ($P < 0.05$, Mann–Whitney–Wilcoxon test). We have now included these observations in the manuscript.

v) That hot spot genes facilitate rapid diversification.

The analysis in Fig. 7 is done per species, so it should be little affected by removing a single species. The analysis of diversity of Fig. 8b (SH test) and 8c (nucleotide diversity) could indeed be affected. Hence, we re-made the analysis on a species and genus basis. The results are shown in Supplementary Dataset 5. The analysis of SH values have very low statistical power on a species per species (or genus) basis and showed no significant results. Regarding the analysis of nucleotide diversity, we show that no single species/genus is driving the results. We have now included these observations in the manuscript (Page 12).

We have made a small reference to this problem in Methods (under "Data"), and direct to the different figures in the text (when each result was presented). We have opted not to discuss this issue in detail because the results are the same and it made the text even heavier.

2) *Some genera are not represented.* We have not excluded species or genera beyond the constraint that we need a minimal number of complete genomes per species to include them. It is true that many phyla are not represented at all because there wasn't (and there still isn't for the majority of them) enough data available (regarding genomes assembled without gaps).

Comment nº 1.3:

The methods are well described and enough details are given. The manuscript is overall well written, but a bit difficult to follow due to the many numbers and statistical tests. Some of the numbers could perhaps be moved to a table and summarized.

Answer to comment nº 1.3:

We have made a significant effort to re-write large parts of the manuscript. We have moved some more numbers and tests to legends, figures or tables. We hope to have improved the text.

Comment nº 1.4:

The results are discussed in the context of the previous literature. However, a discussion about the selective advantage associated with different types of hotspots is lacking – what is the biological significance of the patterns observed? Or are the frequencies and different types of hotspots just an effect of random chance?

Answer to comment nº 1.4:

The section was completely re-written to provide a more complete discussion of the results.

Reviewer 2

I like this paper very much and as expected with these authors, it is a careful and detailed analysis.

We thank the Reviewer for his/her appreciation of our work.

Comment n° 2.1:

My overall comment is that the paper might benefit from shortening and tightening up on the editing. I felt some of the explanations were a little overly long.

Answer to comment n° 2.1:

Following this comment, and the suggestion of Reviewer #1, we have added some more schemas to the main text. This has allowed us to shorten some parts of the text. We have also re-written large parts of the Discussion (Reviewers #1 and #3 asked to add some text, so the net gain there was null).

Comment n° 2.2:

Page 3, line 56: might be also interesting to include the paper from Sharp et al on evolutionary rate variation around the genomes of enterobacteria. It was the first of its kind, i think. (Sharp, P.M., Shields, D.C., Wolfe, K.H. and Li, W.-H. (1989) Chromosomal location and evolutionary rate variation in enterobacterial genes. Science 246, 808-810).

Answer to comment n° 2.2:

Indeed. We have added the reference.

Comment n° 2.3:

Page 5, line 111: spots should be spot.

Answer to comment n° 2.3:

We have corrected it.

Comment n° 2.4:

Page 5, line 114: This is a bit of rhetoric. no need to say "state of the art".

Answer to comment n° 2.4:

We have eliminated the expression 'state-of-the-art'.

Comment n° 2.5:

Page 7, line 158: relatively should be relative.

Answer to comment n° 2.5:

We have corrected it.

Comment n° 2.6:

Page 8, top part of the page. Consider shortening this or tightening the language. I found it very verbose. Would a table help?

Answer to comment n° 2.6:

We have indeed shortened the text (without loss of information). Most of the data is already in Figure 5, so we thought there was no need for a new table.

Comment n° 2.7:

Page 8, line 190: change "[...]and 24% others[...]" to "[...]and another 24%[...]"

Answer to comment n° 2.7:

We have re-written the sentence.

Comment n° 2.8:

Page 10: The beta diversity part is not explained well enough, I feel. Could you add a few sentences, perhaps explaining how this works? I understand beta diversity and I think I understand this analysis, but I am not certain that I understand it, so perhaps a bit of help for the reader might be in order?

Answer to comment n° 2.8:

We have simplified the explanation in the Results section and described the concepts in more detail in the Methods. We hope this part of the text is now clearer.

Comment n° 2.9:

Page 15, line 353: "Expectedly" is a bit awkward. Perhaps "in line with expectations"?

Answer to comment n° 2.9:

This section was extensively re-written and we have eliminated this expression.

Reviewer 3

This study from Oliveira and collaborators aims to identify recombination hotspots in bacterial genomes and to describe their genomic distribution and origins. Authors analyzed genomic data available for 80 bacterial species to infer recent HTG events and identify mobile genetic elements potentially involved in these events.

Although results from this meta-analysis are convincing and may provide interesting insights about the dynamics of bacterial genomes faced to HGT, I don't think that put in a sufficiently broad evolutionary nor functional context. Especially, I found this study very descriptive, both in the results (where authors intentions' are often missing) and the methods, too focused on technical approaches (e.g. statistical measures of HGT) and the discussion too vague about potential evolutionary implications. Especially, this study is mostly focused on genomic aspects (genome size, composition, structure) and MGE dynamics but very few on functional aspects and bacterial ecology (a bit is however mentioned about antibiotic resistance or endosymbiosis L349-350 but without real discussion). I wanted to see more about what can we say about differences observed among species regarding what we now about their ecology, phylogenetic relations, cell cycle, lifestyle?

Furthermore, the way analyses are described makes them very hard to understand at first glance. For instance, most of the very first paragraph (L96-113) from results is confusing without the help from the (very clear!) figure S1; definitions are fuzzy and the corresponding method doesn't help so much (I still didn't get what a "pivot genome" and a "spot pan-genome" are?). I think this is especially a concern for the Nat Com broad audience and scope.

For these reasons I suggest authors to submit their paper maybe to a more specific journal, after deep revision of the manuscript. Please found below more specific comments that I hope will help the authors to improve their work:

We thank the Reviewer for the many suggestions and comments made to our manuscript.

Comment nº 3.1:

ABSTRACT

L30: Please be more clear about the fact that the clustering increases with genome size.

Answer to comment nº 3.1:

Done.

Comment nº 3.2:

L31-33: This sentence is very descriptive and too vague; I know authors tried to summarize very contrasted results but their should put more emphasize on concrete and interesting examples (e.g. the concentration of some classes of MGE in a small number of hotspots) rather than fuzzy generalities (e.g. "local and global chromosomal context").

Answer to comment n° 3.2:

Done.

Comment n° 3.3:

INTRODUCTION

L48-51: the way authors made the transition is weird. Plus, the fact that most of HTGs that are lost are not adaptive doesn't exclude purifying selection.

Answer to comment n° 3.3:

We had no intention of excluding purifying (or positive) selection. We have added a sentence and a reference for this.

Comment n° 3.4:

L72: What kind of structural changes?

Answer to comment n° 3.4:

We have specified them.

Comment n° 3.5:

L76: Define DDE

Answer to comment n° 3.5:

DDE is the name of this family of recombinases. It's not an acronym. It stands for the key residues in the protein.

Comment n° 3.6:

L85-88: This sentence sounds a bit like a circular reasoning, maybe reformulate?

Answer to comment n° 3.6:

We reformulated the text. There are two processes, homologous recombination at core genes and integration of HTgenes. In this precise example, it was an ICE that integrates by site-specific recombination (hence there is no circular reasoning).

Comment n° 3.7:

L89: This sentence alone reveals that the purpose of the study is very vague and not put in a sufficiently broad context (see major comments). Also, precise "recombination" hotspot.

Answer to comment n° 3.7:

The sentence reads "we tested if hotspots exist in a large and diverse panel of bacterial species". There is no vagueness since this is a statistical test. We have nevertheless re-written the sentence to state the relevance of the study.

Comment n° 3.8:

L93: *precise in Bacteria*

Answer to comment n° 3.8:

We have added it.

Comment n° 3.9:

RESULTS

L96: *Provide here an introducing sentence about your purpose with this analysis.*

Answer to comment n° 3.9:

Done.

Comment n° 3.10:

L97: *Provide a supplementary figure explaining how you defined core, pan and accessory genomes (not clear from the text).*

Answer to comment n° 3.10:

We have now added two supplementary figures (Supplementary Fig. 1, 2) explaining how we computed core, accessory, and pan-genomes, and how we defined spots, spots pan-genome, and HTgenes.

Comment n° 3.11:

L100-103: *Incomprehensible without the figure. Plus, what is a pivot genome? the closest from the root of a clade? Or was it arbitrary choose by authors (which is completely fine to me as far as it is clearly indicated). Intervals, spots, breaks points, etc should be more clearly defined directly in the result section.*

Answer to comment n° 3.11:

Pivot. We have now defined pivot more clearly. It's a reference genome that (as indicated in the previous version of the manuscript) typically corresponds to the first sequenced one (which is typically also the type strain of the species).

Intervals, spots, etc. These are described in the Methods. A precise definition in the results would create great redundancy and would make the text even longer (a comment from the other reviewers). We have, however, introduced a schema from Supplementary Fig. 3 in Fig. 1. This was now introduced early in the results. We hope this makes the text clearer.

Comment n° 3.12:

L110-111: *I didn't get what a "spot pan-genome" is, even after reading the method.*

Answer to comment n° 3.12:

The spot pan-genome is the set of gene families present in the intervals included in the spot. We have re-written this part of the text, which was indeed confusing and added a Supplementary Fig. 2 explaining how we defined spot pan-genome.

Comment n° 3.13:

L114: This over-use of compound words doesn't help non-initiated readers.

Answer to comment n° 3.13:

We agree that acronyms make texts heavy. But they are in certain cases necessary to make the text and the definitions precise. Following this and Reviewer's #1 comments we added a table explaining all acronyms (please see Table 1).

Comment n° 3.14:

L116: Could you provide a much less confusing acronym than "HTGs"? E.g. HTgenes

Answer to comment n° 3.14:

We have followed this suggestion.

Comment n° 3.15:

L154: Provide here an introducing sentence about your purpose with this analysis.

Answer to comment n° 3.15:

Done.

Comment n° 3.16:

L197-198: Why this statement?

Answer to comment n° 3.16:

We have changed the text to make the statement clearer.

Comment n° 3.18:

L218-221: The link between the result and the interpretation are not obvious at first glance

Answer to comment n° 3.18:

We have re-written the text.

Comment n° 3.19:

L236-248: unless the scope of the paper is purely technical, this part should be in the method, and only briefly described in results

Answer to comment n° 3.19:

Reviewer #2 asked for more details in this section, whereas Reviewer #3 requests fewer. We have simplified the text in the Results and added more details in the Methods.

Comment n° 3.20:

DISCUSSION:

L288: Author should start the discussion putting they study in a broader context. Plus, the number of hotspots is not informative, first because there's no range across species nor statistics given by authors, second because one has no obvious expectation regarding which number if should be. Why is it surprising? What do we expect?

Answer to comment n° 3.20:

There are several points in this comment:

- "Put the results in a broader context [...] Why is it surprising? What do we expect?". We re-wrote the entire discussion.
- "The number of hotspots is not informative". We disagree. The statistics are made species per species, so the number is informative and allows comparisons between species.
- "First because there's no range across species nor statistics given by authors". We don't understand this criticism. All claims are based on statistical tests. The range across species was already provided in Figure 2 of the previous submission and in the supplementary tables. We now also make most key tests at the level of species and the genera.
- "second because one has no obvious expectation regarding which number if should be". We disagree. The statistical test is designed to give an alpha level of 5%, i.e., given the null hypothesis we expect to find zero hotspots in each species.

Comment n° 3.21:

L291: "specific" rather than "certain"?

Answer to comment n° 3.21:

We have re-written this sentence.

Comment n° 3.22:

L295-299: This part should be removed in results or method.

Answer to comment n° 3.22:

Done.

Comment n° 3.23:

L307: Is it the same result than L289?

Answer to comment n° 3.23:

Yes. The paragraph is intended to discuss it from a different perspective. We have modified the text.

Comment n° 3.24:

L307-310: No real discussion here. I have the feeling that this section is purely filling

Answer to comment n° 3.24:

This is meant to introduce the question of the adaptive value of the genetic information of hotspots. We have shortened according to comment 3.23.

Comment n° 3.25:

L312: please explain more clearly the link you are making between your observation and your interpretation.

Answer to comment n° 3.25:

This was extensively re-written.

Comment n° 3.26:

L314: purifying selection?

Answer to comment n° 3.26:

Indeed. This is now specified.

Comment n° 3.27:

L320-321: Is it so surprising regarding the huge size of these classes of MGE? Intuitively if they are greater, I expect those to be found in a smaller number of hotspots.

Answer to comment n° 3.27:

We don't really see why larger MGEs should necessarily be found in a smaller number of hotspots. Except if there is selection for chromosome organization as we propose. We have tried to make this clearer.

Comment n° 3.28:

L322-326: Very naively, could we imagine that MBE dynamics in these regions is so fast (or under very strong purifying selection) that it cannot be detected at the considered evolutionary scales?

Answer to comment n° 3.28:

This part of the text concerns the absence of MGEs in many hotspots. One could indeed imagine that certain hotspots have a faster turnover (eventually caused by selection) leading to MGEs rapid elimination. If so, these regions should show particularly high genetic diversity. However, our novel analysis shows that hotspots without MAPs do not show higher genetic diversity than the others. We have added this to the text.

Comment n° 3.29:

L348-350: Could authors dig a bit more about the link between hotspot and lifestyle? Is there any analysis that could be done in this way, knowing species lifestyle?

Answer to comment n° 3.29:

There are 80 species in our sample, including free-living bacteria, facultative mutualists, obligatory mutualists, commensals, facultative pathogens, and obligatory pathogens. Obligatory pathogens

and mutualists have few HTgenes, and thus necessarily few hotspots. This is now indicated more explicitly. Dividing the rest of species in terms of lifestyle produces very small subsets that cannot be used for such analyses (one must also control for phylogenetic dependence, which removes even more degrees of freedom to the tests). We will do this in the future as more complete genomes become available. Our past experience is that we need at least around 250 species to make this kind of analysis (accounting for phylogenetic dependence). We checked if adding novel genomes to the analysis would allow reaching this number, but we would still be very far from it (there are not many novel species with enough sequenced genomes). The complete genomes database has re-started to grow a year ago. At this rhythm in two years' time there will be enough data for this kind of analysis.

Comment n° 3.30:

L348-355: It's very unclear what authors try to point out here. That the association between the number of HGT and hotspot frequency/density is just an artifact of HTG number / genome size association? Please clarify.

Answer to comment n° 3.30:

The paragraph was re-written. The goal is to discuss the variation in the number of hotspots and its possible determinants.

Comment n° 3.31:

L363: "certain organizational pattern" is too vague.

Answer to comment n° 3.31:

We have now specified this.

Comment n° 3.32:

L364: again, filling sentence.

Answer to comment n° 3.32:

We completed the sentence to show how it links the two parts of the paragraph.

Comment n° 3.33:

L370: It could be interesting to use predicted DNA folding sites, if available in bacteria studied by authors, and look if these sites correlate with hotspot distribution in these species.

Answer to comment n° 3.33:

Unfortunately the only methods we know for bacteria only predict local properties of the DNA not the large topological domains we are mentioning here. This will be doable once enough HiC data becomes available.

Comment n° 3.34:

L374: "certain function", too vague.

Answer to comment n° 3.34:

The sentence was deleted. We discuss hypothesis of why hotspots tend to be closer to core genes with functions involved in repair, regulation, and recombination elsewhere in the discussion (note that these functions are depicted in Fig. 3).

Comment n° 3.35:

L376-377: What is the link with the previous sentence and what is the authors' point here?

Answer to comment n° 3.35:

The sentences were re-written.

Comment n° 3.36:

METHOD:

L383-384: It could be interesting to look backward if hotspots are enriched in these CDS.

Answer to comment n° 3.36:

Indeed, one expects these regions to over-represent pseudogenes (because they over-represent HGT). The problem is that GenBank files treat pseudogenes in a very heterogeneous way (they are not annotated in many genomes), so putting these pseudogenes back into the analysis will not give reliable results. To analyze this accurately, one needs to identify pseudogenes in all the genomes using the same methodology. We think this is out of the scope of the present work.

Comment n° 3.37:

L418-419: Does it mean that potential HTG that led to these rearrangements were not estimated?

Answer to comment n° 3.37:

Indeed. As stated in the previous version of the manuscript this forces us to ignore these genes. They make 2% of all HGT. We have made this clearer.

Comment n° 3.38:

L423-438: This full section is unclear and should be rewritten.

Answer to comment n° 3.38:

We have made a number of changes to make it clearer, including giving options of each program, adding some explanations, and adding a new Supplementary Fig. 2. We hope it's clearer.

Comment n° 3.39:

L441: evolutionary dynamics?

Answer to comment n° 3.39:

We have added the word "evolutionary" as suggested.

Comment n° 3.40:

L459: any reference for this?

Answer to comment n° 3.40:

This is a classical null hypothesis (when we don't know, we say it's random for parsimony). We state it for mathematical clarity. If the Reviewer thinks it's important to add other constraints to make the model more complex, we will add them.

Comment n° 3.41:

L460-461: is this proportion is highly variable among species? If so, could it impact your results?

Answer to comment n° 3.41:

According to the original publications this is not highly variable among species. For example, the estimate for the number of genes in operons found in Zheng, Genome Research, 02 is (our calculation) 3.15 ± 0.06 for a set of 42 genomes (coefficient of variation=0.1). This was now added to the text.

Comment n° 3.42:

FIGURES:

Fig1: *sulcia* is a species name, *Buchnera* a genus, please uniformize.

Answer to comment n° 3.42:

Sulcia is not a species name. It is a genus name. It is often written as *Candidatus Sulcia*, because the genus has not yet been officially accepted (hence, the "*Candidatus*" as is standard in systematics). We have now added Ca. in the figure to indicate this.

Comment n° 3.43:

Fig2a: very descriptive panel. What can we conclude from it? Is there any family effect? Lifestyle?

Answer to comment n° 3.43:

Most of our work focuses on summary statistics. We believe it is important to show the dispersion of the most important results. That's the goal of this figure. We can remove it for supplementary material if this fits better the editorial policy of Nature Communications.

Comment n° 3.44:

Fig3a: what the letter at the bottom stand for?

Answer to comment n° 3.44:

They represent the names of the classes of non-orthologous groups (NOGs). We have now made this clearer in the figure legend.

Reviewer 2

This is an exciting paper that provides su with some very interesting and detailed insights into the process of horizontal gene transfer in prokaryotes. The current version is much improved in terms of readability and I have no problems with it. There are still some grammatical and language issues, but they are fairly minor.

We thank the Reviewer for once again recognizing the interest and potential impact of our work in the field.

Comment nº 2.1:

There are still some grammatical and language issues, but they are fairly minor.

Answer to comment nº 2.1:

We went over the manuscript and corrected some language issues. Our changes were made in order to simplify the text, remove redundancies, and make better transitions between paragraphs in the discussion. We underline that we have not added, removed or changed any result, and we have not changed any significant conclusion.

Reviewer 3

Authors greatly improved they manuscript and clarified most of my queries.

We thank the Reviewer once again for his/her suggestions on our manuscript.

Comment n° 3.1:

L90-92: This sentence should be re-written, because the causal link between the observation and the proposed mechanism is not intuitive.

Answer to comment n° 3.1:

We have re-written the sentence.

Comment n° 3.2:

L206: Very naively, the observed trade-off between prophages and integrative elements occurrence in hotspot couldn't be simply due to their large size, and so their potentially large negative effect on fitness? I guess that in the same way, it is also rare to observe several prophages or several ICEs in the same hotspot?

Answer to comment n° 3.2:

We believe this results from a misunderstanding of the text. The correlation is between elements in the same spot, but not necessarily in the same interval (actually this is quite rare). The size of the element is thus without much relevance. Also, we note that ICEs and prophages have similar sizes and there is thus no reason that prophages and prophages (and ICEs and ICEs) could co-occur but not ICEs and prophages. We have changed the text to make this clearer.

Comment n° 3.3:

L250-252: Be more specific about "high genetic diversity".

Answer to comment n° 3.3:

We have clarified this point.

Comment n° 3.4:

L284-293: The discussion starts too abruptly without general context but with method elements. Maybe the authors should remove the very first paragraph from the discussion, or to summarize it in a single sentence, while putting less (or no) emphasize on the method they used.

Answer to comment n° 3.4:

Some of these initial changes in the Discussion section were an attempt to put the study in a broader context, as asked by Reviewer 3 in its previous comment 3.20. Yet, after reading our changes, we agree that this section needed some re-writing to be of better style. We have

eliminated this first paragraph and changed the beginning and end of several paragraphs of the discussion to improve flow.